# Quantum fluctuations drive nonmonotonic correlations in a qubit lattice

**Alejandro Lopez-Bezanilla**[1] ✉**, Andrew D. King** [2]**, Cristiano Nisoli**[1] **& Avadh Saxena**[1]

Fluctuations may induce the degradation of order by overcoming ordering interactions, consequently leading to an increase of entropy. This is particularly evident in magnetic systems characterized by nontrivial, constrained disorder, where thermal or quantum fluctuations can yield counterintuitive forms of ordering. Using the proven efficiency of quantum annealers as programmable spin system simulators, we present a study based on entropy postulates and experiments on a platform of programmable superconducting qubits to show that a low level of uncertainty can promote ordering in a system impacted by both thermal and quantum fluctuations. A set of experiments is proposed on a lattice of interacting qubits arranged in a triangular geometry with precisely controlled disorder, effective temperature, and quantum fluctuations. Our results demonstrate the creation of ordered ferrimagnetic and layered anisotropic disordered phases, displaying characteristics akin to the elegant order-by-disorder phenomenon. Extensive experimental evidence is provided for the role of quantum fluctuations in lowering the total energy of the system by increasing entropy and defect clustering. Our thorough and comprehensive application of an intentionally introduced noise on a quantum platform provides insight into the dynamics of defects and fluctuations in quantum devices, which may help to reduce the cost associated with quantum processing.

Entropy-driven ordering mechanisms have been successfully employed to explain the formation of colloid materials[1,2], although only a few observations in macroscopic systems[3] have been reported thus far. Experimental investigations into magnetic ordering driven by thermal and quantum fluctuations have been limited, despite the potential impact a deep understanding of the fundamental mechanisms governing the formation of ordered structures could have on the assembly and organization of magnetic nanostructures.

The state of a disordered system is defined by the even or uniform distribution of its constituent elements throughout space. Conversely, the concept of order pertains to the arrangement of these elements, where they tend to segregate or group together based on their similar properties. The second law of thermodynamics established entropy as a measure of disorder, with Boltzmann's expression for the entropy of a closed system, $S \sim \ln \Omega$, stating that the entropy $S$ increases with a higher number of accessible states $\Omega$ for the system[4]. Temperature, which relates to thermal fluctuations, plays a crucial role in altering the entropy or degree of ordering in a system. Higher temperatures provide more ways to distribute energy among the available states, leading to higher entropy values. While it may seem counterintuitive, an increase in thermal fluctuations can actually promote ordering within a system through the phenomenon known as order by disorder[5–8]. This phenomenon occurs in frustrated magnetic systems, where the classical ground state manifold possesses higher symmetry than the underlying Hamiltonian. Both thermal and

[1]Theoretical Division, Los Alamos National Laboratory, Los Alamos 87545 NM, USA. [2]D-Wave Systems, V5G, Burnaby, BC, Canada.
✉e-mail: alejandrolb@gmail.com

quantum fluctuations can break the degeneracy and induce ordering in the system while increasing its entropy.

In the present investigation, we design magnetic systems with a highly frustrated classical ground state manifold in which to observe an increase of correlations between magnetic moments when thermal and quantum fluctuations are finely tuned. The idea that adding quantum fluctuations could promote order appears counterintuitive. For example, in magnetic kagome lattices a monotonic increase of magnetic moment correlations is found with increasing temperature. Here we demonstrate that the inclusion of quantum fluctuations in asymmetric (Lieb) kagome lattices can produce nonmonotonic correlations among magnetic moments.

## Results and discussion

This challenging problem for established macroscopic magnetic realizations[9–13] can be implemented in a straightforward manner in a quantum annealer hardware. Free of spurious interactions and defects that plague spin ice systems, the platform of interconnected qubits provided by D-Wave Quantum (Fig. 1a) enables the observation of many-body interactions driven uniquely by entropic effects. By using entanglement and superposition of tailor-designed quantum states, we unveil with a high degree of accuracy the interplay between temperature and domain-wall correlation in a continuously distorted while frustrated qubit lattice. In particular, the maximally frustrated kagome lattice is taken as a reference, and an effective deformation is introduced by modifying coupling parameters that mimics an applied strain to the real-space lattice. The kagome lattice is thus placed in a broader antiferromagnet geometry, showing that its constrained and correlated disorder is a critical point between fully ordered and uncorrelated disordered states. This flexibility of the degree of disorder in one single topology allows us to study the nonmonotonic evolution of qubit-qubit ordering within an effective range of temperature.

Our model of lattice transformation is described as the continuous deformation of a kagome (depleted triangular lattice) to a Lieb lattice (depleted square lattice), and to a deformed hexagonal lattice, as shown in Fig. 1c−e. Each site has a classical representation given by a magnetic moment in either $\langle\uparrow\rangle$ or $\langle\downarrow\rangle$ state orientation. In the

following, we will refer to two types of sites, $\sigma$ and $\Sigma$, denoting the Ising qubits circled in black and yellow, respectively. The unit cell in Fig. 1b features three nodes and two antiferromagnetic coupling constants $J$ and $J_L$. Different classical ground states can be obtained depending on the ratio $J_L/J$, which controls the system frustration.

$J_L/J < 1$ yields an ordered ferrimagnetic configuration (Fig. 1c), where all $\sigma$ qubits have the same alignment, and opposite to that of $\Sigma$ qubits. The net magnetic moment per unit cell is $|M| = 1/3$. If $J_L/J > 1$, the weaker coupler $J$ is frustrated in a layered ground state. $\sigma$-site qubits align antiferromagnetically (Fig. 1e) along the stronger (shorter in the lattice representation) bond $J_L$, forming layered, parallel antiferromagnetic lines, as shown by the solid gray lines in Fig. 1e. The $\Sigma$-site qubits can be viewed as buffers that do not transfer information between antiferromagnetic lines, namely the energy of the ground state is unaltered upon a spin-flip operation on any of those sites. They are shown in white since they exhibit no specific order. The ordered lines are thus mutually uncorrelated and disordered in the vertical direction. This is only true in the ground state, as we shall see below, much as in the prototypical domino model of order by disorder[5]. Fluctuations develop correlations among antiferromagnetic lines, making the layered phase a quasi-one-dimensional (Q1D) phase, and the average magnetization zero.

At the critical point, $J_L = J$, the lattice becomes a frustrated Ising antiferromagnet on a kagome lattice[14,15] (Fig. 1d). The low-energy state manifold of the two models consists of disordered qubits that obey the ice rule, namely the three spins in a triangle must add up to $\pm 1$. In the ground state at zero temperature qubits are disordered and strongly correlated, with a finite correlation length[16]. The average magnetization is zero.

Note that both the ferrimagnetic ($J_L/J < 1$) and the layered ($J_L/J > 1$) phases are deviation from the kagome ice configuration ($J_L/J = 1$), as both obey the ice rule as defined above in terms of minimal $\pm 1$ charge[16]. Therefore, at $T = 0$ the hexagonal qubit ice is the structural critical point between an ordered ferrimagnetic phase and an anisotropically disordered layered phase. Analogously, square ice had been also shown to be a structural critical point between antiferromagnetic order and a state of subextensive disorder[15,17,18]. After discussing the

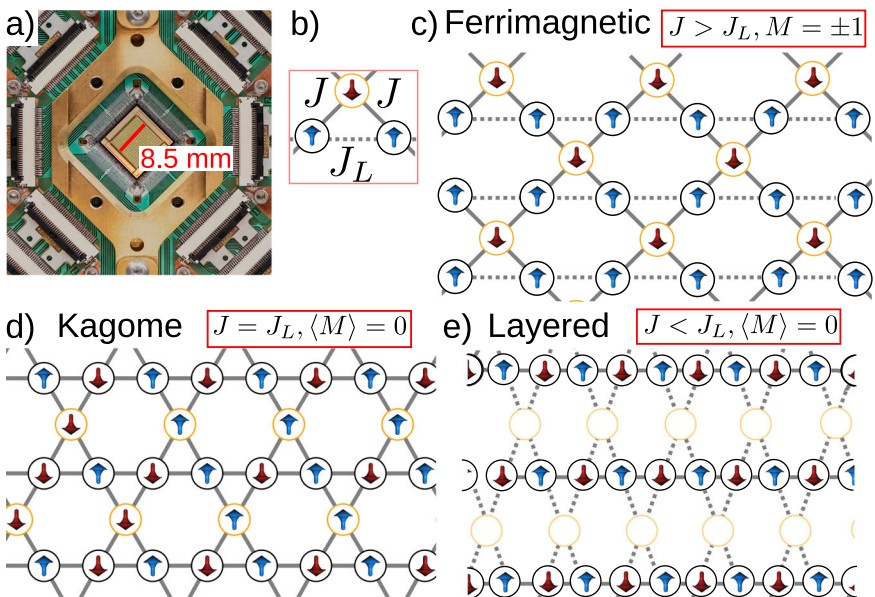

**Fig. 1 | The system under study in its three ground states. a** The D-Wave quantum processing unit in use. (b) Unit cell of studied model composed of three nodes coupled in a triangle with two different couplers, $J$ and $J_L$. Panels (**c**−**e**) show the Lieb lattice with anisotropic couplings $J$ (solid gray lines), and $J_L$ (dotted gray lines). Solid gray lines represent stronger couplings. Topological Lieb-to-kagome

transformation is equivalent to reducing $J_L$ with respect to $J$ that is kept constant. In (**c**), $J > J_L$ yields a ferrimagnetic ground state configuration. The kagome ice configuration in (**d**) is reached when $J_L = J$. The layered configuration in (**e**) is equivalent to considering $J < J_L$. Up and down magnetic moments are depicted in blue and red respectively.

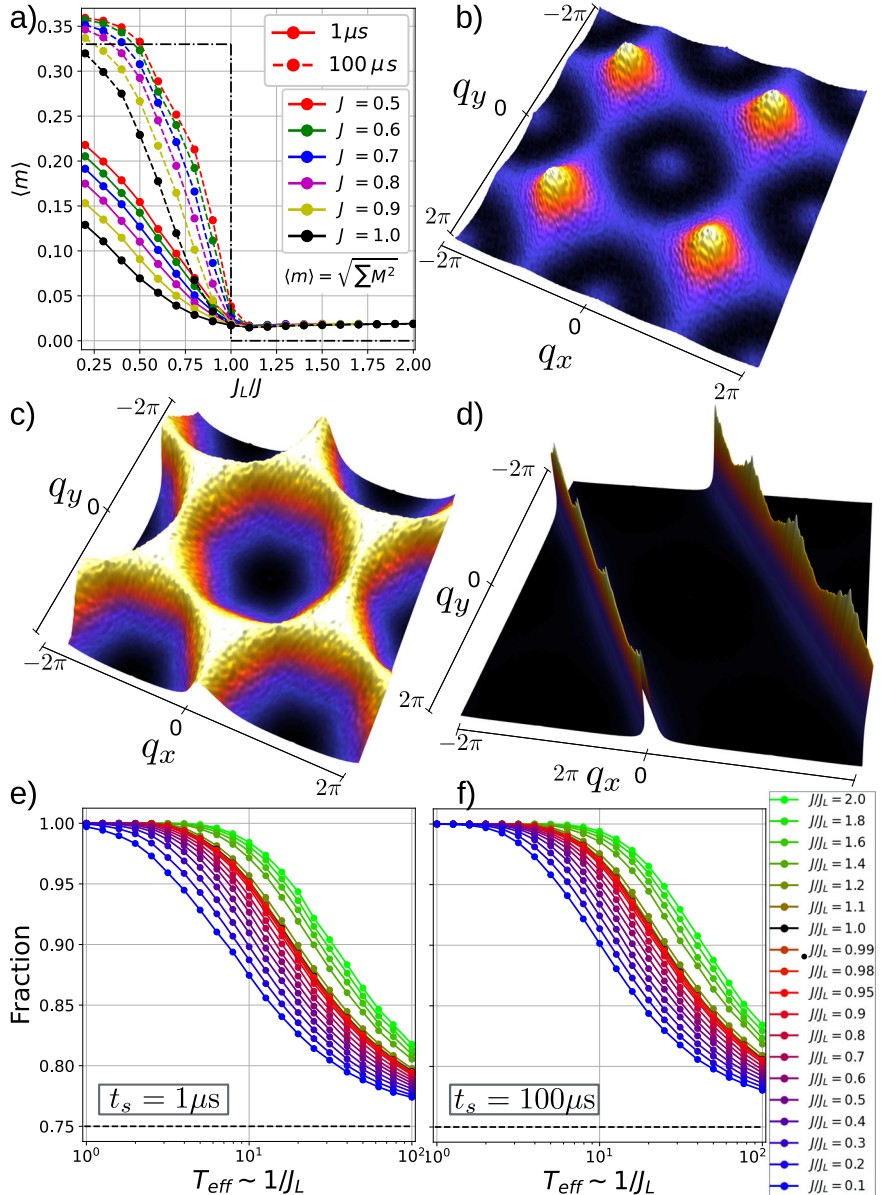

**Fig. 2 | Structure factors of three ground states obtained from quantum annealing experiments. a** Average magnetization per qubit $\langle m \rangle$ vs. $J_L/J$ obtained from quantum annealing for different values of $J$ and therefore different effective temperatures, $T_{eff}$. In the ground states $|M| > 0$ only for $J_L/J < 1$. The various ground state symmetries are shown via the Fourier transform of the qubit-qubit correlation after annealing for $J/J_L = 1.5, 1.0, 0.75$. **b** corresponds to the ferrimagnetic case, (**c**) to the kagome ice, and (**d**) the layered case. All phases obey the ice rule, as shown in plots of fraction of ice-rule obeying vertices vs. $T_{eff}$ for fast (**e**) and slow (**f**) annealings.

anticipated ground state, we move on to investigating a finite-sized realization of $J_L < J$ kagome-derived lattices, where the system dynamics is dominated by quantum rather than thermal fluctuations.

Our finite-size qubit model consists of a lattice of $21 \times 21$ anti-ferromagnetic (AFM) chains, along with the mediating qubits, resulting in a total of 641 logical qubits. To represent a logical spin, three qubits are connected by strong ferromagnetic couplings with a magnitude of $J = -2$, resulting in a total of 1923 qubits embedded in a D-Wave Pegasus topology. The collective dynamics of these qubits is described by a quantum transverse-field Ising model, governed by the Hamiltonian:

$$H = \mathcal{J}(s) \left( J_L \sum_{\langle \sigma, \sigma' \rangle} \hat{\sigma}^z \hat{\sigma}'^z + J \sum_{\langle \sigma, \Sigma \rangle} \hat{\sigma}^z \hat{\Sigma}^z \right) - \Gamma(s) \left( \sum_{\sigma} \hat{\sigma}^x + \sum_{\Sigma} \hat{\Sigma}^x \right) \quad (1)$$

where $\hat{\sigma}, \hat{\Sigma}$ are Pauli matrices of the corresponding qubits on $\sigma, \Sigma$ sites, respectively. In the absence of the transverse field, $\Gamma = 0$, Fock products of eigenstates of $\sigma^z$ are eigenvalues of the Hamiltonian, and the problem maps exactly onto the classical one described above. If $\Gamma \neq 0$, the terms within the first parenthesis do not commute with the terms within the second parenthesis, and the transverse field subjects the spin degrees of freedom to quantum fluctuations. The unitless parameter $s = t/t_f$ controls the annealing progress from the quantum-mechanical superpositions of states at $s = 0$ to the classical state at $s = 1$. The Ising energy scale is thus controlled by $\mathcal{J}(s)$ which increases as the quantum fluctuations $\Gamma(s)$ away out according to well-established annealing protocols.

Starting from a transverse-field induced quantum-superposition state, the forward annealing protocol is used to obtain thousands of classical configurations at different $J_L/J$ ratios. Figure 2a shows the magnetization per qubit vs. $J_L/J$ for a series of $J$ values, averaged over

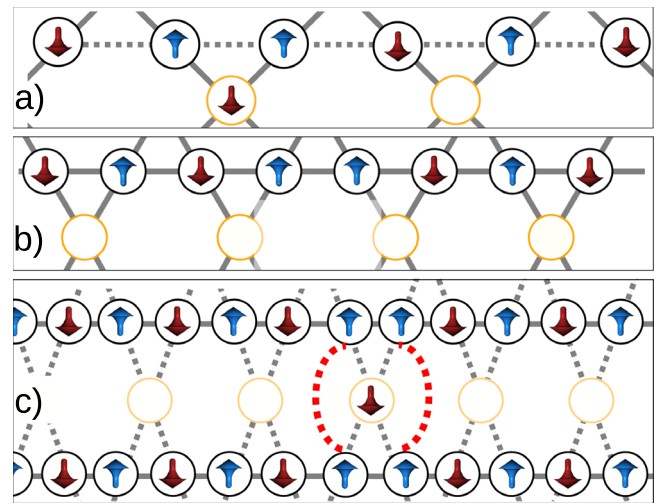

**Fig. 3 | Entropy-induced correlations of the domain walls. a** A domain wall sitting on top of a $\Sigma$ site fixes the qubit with an entropic cost that is lower in energy than in the case shown in (**b**). In (**c**), two $\sigma$-site domain walls are "paired" if both sit around the same $\Sigma$ site. Pairing creates entropic advantage with respect to two domain walls away from each other by releasing a $\Sigma$ floppy qubit. Dotted red lines point out the transverse correlation between the domain walls. Up and down magnetic moments are depicted in blue and red, respectively.

25600 annealed samples. In the ground state of an infinite lattice in the ferrimagnetic phase ($J_L/J < 1$), where the magnetization is the order parameter, the two twofold coordinated spins antialign with the spin in the fourfold vertex. The stepwise dashed-dotted line points out the absolute value of its magnetization at $T = 0$, with a constant value of $1/3$ for $J_L/J < 1$ and of zero for $J_L \geq J$. The open boundaries in the chains induce fluctuations that cause the initially flat line to transform into a smoothly increasing curve for $J_L/J > 1$. The finite value of $J$ precludes the magnetization $\langle m \rangle$ from abruptly transitioning to $1/3$, allowing for the possibility of deviations from the 2-up-1-down rule in the ferrimagnetic phase. At the exact value of $J = J_L$, the ensemble meets the degenerate ice-rule ground states of the kagome lattice, with a $\pm 1$ net magnetization on each triangle realizing an overall $\langle m \rangle = 0$. The magnetization of the slab acquires finite values due to its finite size. The quality of the annealing process is reflected in the two sets of lines plotted in Fig. 2e, f. A swift annealing of $t_f = 1\mu s$ demonstrates to be less efficient than a slower $t_f = 100$ $\mu s$ annealing yielding lower $\langle m \rangle$ in the ferrimagnetic phase.

Structure factors indicating the underlying spin alignment for each $J_L/J$ ratio are displayed in Fig. 2b-d. $S(\mathbf{q}) \sim \sum_{i,j} e^{-\mathbf{q}(\mathbf{r}_i - \mathbf{r}_j)} \langle \mathbf{s}_i \cdot \mathbf{s}_j \rangle$ is the Fourier transform of the correlation between the $i$th- and the $j$th-spin, and pinpoints a number of features: the ferrimagnetic order is clearly visible in Fig. 2b with peaks at $\mathbf{q} = (\pm \pi, \pm \pi)$ of the Brillouin zone. The $S(\mathbf{q}) \neq 0$ at $\mathbf{q} = (0, 0)$ shows evidence of long-range correlations. Figure 2d shows a ridged line structure expected in the layered phase as corresponds to the one-dimensional antiferromagnetic order. Figure 2c shows the smeared hexagonal pattern typical of the kagome antiferromagnet.

It is also possible to explore higher energy states for fixed frustration $J_L/J$, by varying both $J_L$ and $J$ at a constant ratio. The coupling strength is subjected to at-will variations to mimic the effect of temperature, so that increasing $J$ is equivalent to reducing the effect of thermal fluctuations. With discrete variations of the coupling strength ($J_L = 0.1 .. 1.0$ in steps of $0.1$) the system's effective temperature is mimicked, and it can be regarded as inversely proportional to the coupling strength between logical qubits. In this scenario, $T_{\text{eff}} = 1/J_L$ has been used[15] to represent a useful notion of effective temperature, $T_{\text{eff}}$. Figure 2d, e display the fraction of triangles obeying the ice rule plotted as a function

of $T_{\text{eff}} = 1/J_L$, for fixed values of $J_L/J$ during fast (d) and slow (e) annealings. The system obeys the ice rule at low $T_{\text{eff}}$.

The dimensionally reduced phase affords interesting similarities with the domino model[5], where correlations develop as the system is subject to thermal fluctuations. The elegant order-by-disorder mechanism is due to the presence of excitations, absent in the ground state. Other layered nanomagnets recently realized[19,20] behave similarly by promoting linear arrangements that are mutually uncorrelated in the ground state, but can be correlated by entropy under stress. Interestingly, an analogous mechanism of entropy-induced correlations in the Q1D phase can also be expected, which relies on the sea of free $\Sigma$ spins, that are uncorrelated in the ground state but correlate under fluctuating excitations.

Figure 3 illustrates that this mechanism proceeds through an entropic gain stemming from the correlation of domain walls. It is essential to note that in the ground state of the quasi-one-dimensional (Q1D) system, all $\sigma$-qubits lines (indicated by black circles) exhibit antiferromagnetic ordering, while the $\Sigma$ qubits (indicated by yellow circles) have random orientations because their net coupling with the antiferromagnetic $\sigma$ spins is zero. Consequently, in the ground state, there is no correlation among the antiferromagnetic lines. Excited states, on the other hand, involve the formation of domain walls on these lines, which separate domains with different antiferromagnetic orientations.

It is important to observe that a domain wall in an antiferromagnetic line possesses lower energy when located at the position of a $\Sigma$ qubit, effectively locking it in an orientation opposite to that of the domain wall. In this scenario, the energy cost of creating a domain wall between two contiguous $\sigma$ sites at both sides of a $\Sigma$ spin is $\Delta E = 2(J_L - J)$. In contrast, placing the same domain wall away from a $\Sigma$ spin (as depicted in Fig. 3b) would result in a higher energy cost, $\Delta E = 2J_L$. It is therefore energetically advantageous for a domain wall to be adjacent to a $\Sigma$ spin.

Next, consider two domain walls on two adjacent antiferromagnetic lines, with $\Sigma$ spins in between. If these two domain walls lock onto the same $\Sigma$ spin, they, and the domain they separate, adopt the same orientation, thereby contributing to a transverse correlation. However, there is no energy preference for having the two domain walls aligned by the same $\Sigma$ spin. The energy remains the same if they lock onto two different $\Sigma$ qubits. Nevertheless, there is an entropic advantage in having the two domain walls aligned because it allows one $\Sigma$ spin to remain free to fluctuate, resulting in an energy gain of $\Delta S = \ln 2$. Consequently, we can deduce that entropy promotes the alignment of domain walls in parallel chains and favors the existence of floppy spins, increasing the transverse correlation among these lines.

To investigate this fluctuation-induced transverse correlation, both perpendicular across lines, $\langle \sigma \sigma' \rangle_\perp$, and parallel along a line, $\langle \sigma \sigma' \rangle_\parallel$, nearest-neighbor correlations are considered and defined as

$$\langle \sigma \sigma' \rangle_\perp = \left\langle \sum_{l,i}^{*} \sigma_i^{(l)} \sigma_i^{(l+1)} \right\rangle \tag{2}$$

$$\langle \sigma \sigma' \rangle_\parallel = \left\langle \sum_{l,i}^{*} \sigma_i^{(l)} \sigma_{i+1}^{(l)} \right\rangle \tag{3}$$

where the star $^{*}$ denotes the normalized sum over $\sigma$ spins at the $i$th position along the $l$th line. $\langle \rangle$ denotes the average over 25600 classical states. At $T = 0$, $\langle \sigma \sigma' \rangle_\perp = 0$, since the prevalent energy is the coupling term that antiferromagnetically aligns the mutually uncorrelated chains preventing the formation of domain walls. At large temperature, $J$ is overwhelmed by thermal fluctuations and the whole system becomes uncorrelated. As a result, it is anticipated that $\langle \sigma \sigma' \rangle_\perp$ will exhibit a non-monotonic behavior with respect to the (effective) temperature.

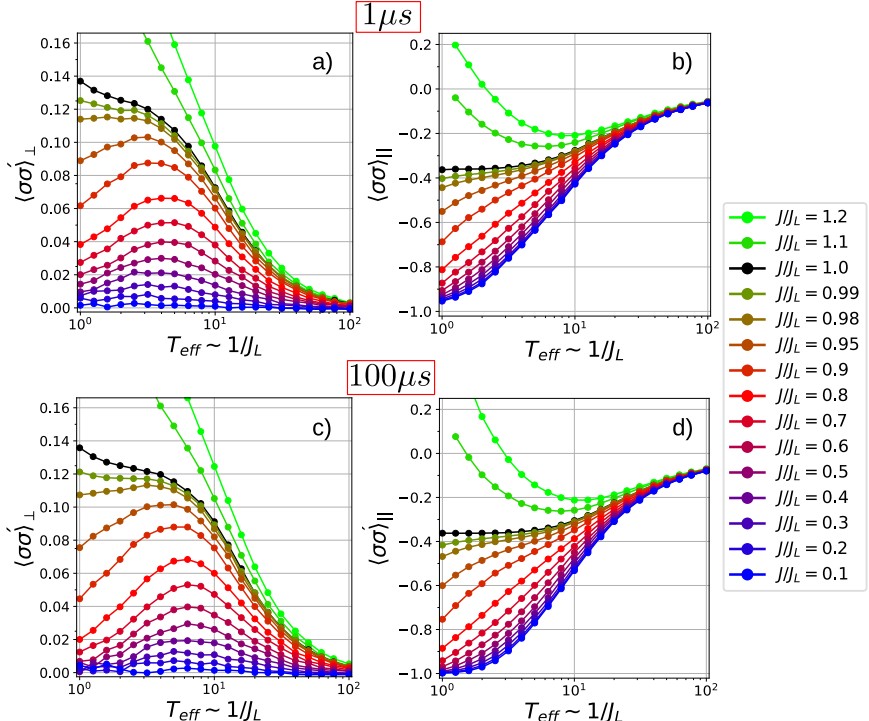

**Fig. 4 | Entropic transverse correlations. a** The transverse correlation $\langle\sigma\sigma'\rangle_\perp$ vs. $T_{\text{eff}} \sim 1/J_L$ showing its nonmonotonic behavior. Curves obtained after annealing time of $t_f = 1\mu s$. **b** Longitudinal correlation along the antiferromagnetic lines. **c, d** Same as in (**a**) and (**b**), respectively, for an annealing time of $t_f = 100\ \mu s$.

In the following, the effective temperature of the parallel chain system will be taken as inversely proportional to the coupling strength between logical qubits, $T_{\text{eff}} \sim 1/J_L$[15]. Figure 4b, d shows $\langle\sigma\sigma'\rangle_\parallel$ plotted as a function of $T_{\text{eff}}$ for a series of values of $J_L/J$. Annealing procedures are set to last $t_f = 1\mu s$ and $t_f = 100\ \mu s$, respectively. As it is generally the case for correlations, the curves are monotonic in $T_{\text{eff}}$ for $J/J_L < 1$ (i.e. when the system is in the Q1D phase). At low-$T_{\text{eff}}$, assuming that the frustration quantifier $J/J_L$ remains close to 1, an antiferromagnetic state ($\langle\sigma\sigma'\rangle_\parallel = 1$) is reached at the lowest $T_{\text{eff}}$. If $J/J_L \to 1^-$, the curves approach uniformly the kagome ice behavior (black line). The collinear correlation for kagome ice at low-$T_{\text{eff}}$ is only slightly higher than the value $\langle\sigma\sigma'\rangle_\parallel = -4/6 + 2/6 = -1/3$, which can be readily obtained via a counting argument on an ice-rule obeying triangle. Curves for $J/J_L > 1$ correspond to a ferrimagnetic regime whose correlations become positive at low-$T_{\text{eff}}$. Linear correlations are slightly stronger for slower annealings as generally expected in optimal an annealing process.

We analyze the transverse correlations $\langle\sigma\sigma'\rangle_\perp$, plotted in Figure 4a, c, which exhibit a nonmonotonic behavior for $J/J_L < 1$. As the value of $J/J_L$ increases, the correlations become stronger at the same effective temperature due to the influence of the $\Sigma$ qubits, whose coupling with $\sigma$ qubits is $J$. For each curve, the temperature associated with the maximum transverse correlation decreases as $J/J_L \to 1$, indicating the transition to an isotropically correlated kagome ice state. In this state, the correlations are monotonic in temperature, with the maximum occurring at the ground state.

The obtained correlation at low $T_{\text{eff}}$, $\langle\sigma\sigma'\rangle_\perp \approx 0.14$, agrees well with experimental findings for kagome systems discussed in refs. [21,22] for a nanomagnetic, artificial kagome annealed via AC demagnetization. The observed correlations, consistently weaker than those of kagome ice, suggest the absence of symmetry breaking in a large system. The extraction of an order parameter reveals at most very small values of around 18% which are explainable with counting arguments for a finite system made of an odd number of lines.

Notably, the transverse correlations shown in Fig. 4a, c exhibit stronger values during faster annealings. While faster annealings

demonstrate only slightly smaller collinear correlations, they display significantly larger transverse correlations. For example, in the case of $J/J_L = 0.9$ at $T_{\text{eff}} = 1$, the faster annealing exhibits an approximately 50% increase in transverse correlation compared to the slower annealing, while experiencing only a 7% decrease in collinear correlation. This suggests that the enhancement of transverse correlation during faster annealings arises primarily from quantum fluctuations rather than from a large number of domain walls.

In the presence of a small transverse field $\Gamma$, the system experiences quantum fluctuations, leading to increased transverse correlations. The second term of the Hamiltonian in Equation (1) can be considered as a perturbative term that breaks the degeneracy of the classical ground state manifold. Configurations with more unlocked qubits, which correspond to more paired domain walls, have lower total energy[23,24]. Therefore, the full quantum Hamiltonian including the transverse field $\Gamma$ generates configurations that, with more paired domain walls and fewer locked $\Sigma$ qubits, are energetically favored. Although $\Gamma = 0$ always at the end of an annealing, decreasing it faster can promote states that are energetically advantageous when $\Gamma \neq 0$. This explains the larger transverse correlation observed during faster annealings.

A comparison between the $\langle\sigma\sigma'\rangle_\perp$ and $\langle\sigma\sigma'\rangle_\parallel$ panels of Figure 4 reveals their opposite evolution at low $T_{\text{eff}}$. This can be attributed to the increased formation of domain walls, leading to a decrease in longitudinal correlation and an increase in transverse correlation through the mechanism of domain wall pairing described earlier. In the ferrimagnetic phase, $\langle\sigma\sigma'\rangle_\parallel$ is equal to 1, while in the layered phase it is equal to $-1$, due to the finite size of the sample.

Subtle variations in the annealing protocols can have significant effects on quantifying the degree of order, as quantum fluctuations play a crucial role in enhancing the spin-spin correlation represented by $\langle\sigma\sigma'\rangle_\perp$. The slow forward annealing protocol used so far anneals an open quantum system that drops out of thermal equilibrium as quantum fluctuations ($\Gamma(s)$) decrease, causing the system response time to exceed the annealing time. By introducing a pause-and-quench

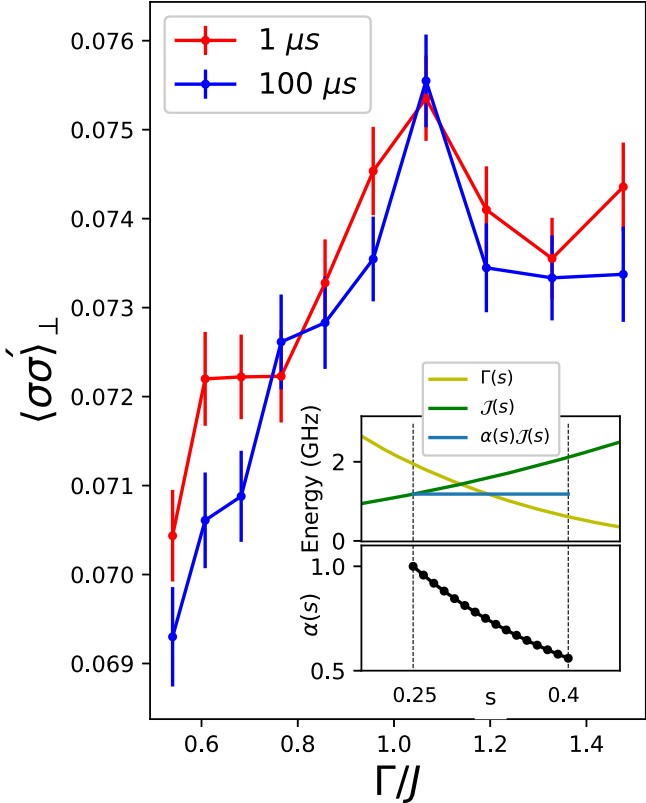

**Fig. 5 | s-Dependent transverse correlations.** Correlations are probed at individual $s$ values using a pause/quench protocol. To eliminate dependence of $\Gamma(s)/J(s)$ on $s$ over the region of measurement, a prefactor $\alpha(s)$ is introduced. Inset shows the flat energy range obtained by multiplying $\mathcal{J}(s)$ by the set of $\alpha$ values shown in the black curve. As a function of $\Gamma(s)/J$, we see a peak in transverse correlations $\langle\sigma\sigma'\rangle_\perp$. This is consistent with a region of order between the perturbative regime ($\Gamma \ll J$) and the paramagnetic regime ($\Gamma \gg J$).

protocol, we can probe midanneal properties: pausing at an $s^* < 1$ allows the system to approach thermal equilibrium of the open quantum system. After the pause, rapidly quenching $s$ to 1 collapses the wavefunction, approximately reading out the system at $s^*$. Note that a change of $s^*$ changes both $\Gamma(s^*)$ and $\mathcal{J}(s^*)$, when the goal is to change only $\Gamma(s^*)$. To achieve that, the couplings $J_{ij}$ are multiplied by a prefactor $\alpha(s^*)$ such that the product $\alpha(s^*)J(s^*)$ remains constant across the range of $s^*$ values to be probed.

Considering the terms proportional to $\Gamma$ in the Hamiltonian of Eq. (1) as the interaction Hamiltonian, a perturbative treatment at first order shows that configurations with the maximum number of floppy qubits are energetically favored. In other words, a small $\Gamma$ field causes a fine splitting of the degeneracy of the classical ground state into a number of energy levels, favoring configurations with a larger number of floppy spins. When $\Gamma$ is annealed very slowly, this mechanism is less relevant although significant than in a swift annealing dynamics where it resembles more a quench on the $\Gamma$ field.

Figure 5 displays the dependence of $\langle\sigma\sigma'\rangle_\perp$ with the degree by which the system is affected by quantum fluctuations induced in the lattice according to a reverse annealing protocol. The figure plots $\langle\sigma\sigma'\rangle_\parallel$ vs. $\Gamma/J_L$, which represents a good quantifier for the amplitude of quantum fluctuations induced in the system. The plot clearly illustrates that the transverse correlation increases with the ratio of the transverse field strength $\Gamma$ to the exchange coupling $J_L$: Larger quantum fluctuations lead to stronger correlations. The curve reaches a maximum value, and for larger $\Gamma$, the transverse Ising models are known to undergo a transition to a quantum paramagnetic phase, typically

occurring at $\Gamma \approx J$, as shown in the figure. This behavior serves as a significant signature of observable quantum effects arising from quantum fluctuations, as it provides experimental evidence of the ability of quantum fluctuations to induce correlations among the chains that are absent otherwise. When considering the same number of defects, those located at both sides of a locked qubit exhibit a lower configuration energy compared to the defects situated in different regions of the parallel chains that involve two intermediate qubits (Fig. 3c).

In summary, we have conducted a series of experiments in a frustrated magnetic lattice to provide understanding of the ability of quantum fluctuations to enhance ordering. Based on fundamental entropic principles, we have demonstrated that ground-state modification by quantum-mechanical fluctuations may induce a domain-wall entropic-interaction mechanism that reduces the disorder in the perpendicular direction leading to nonmonotonic transverse correlations of antiferromangetically aligned magnetic moments. We have demonstrated the continuous deformation of a kagome lattice to ferrimagnetic and layered structures, by engineering the ratio between coupling constants. This flexibility in controlling the degree of disorder within a single topology has enabled us to explore the nonmonotonic evolution of qubit-qubit ordering within an effective range of temperature, to find that a given amount of uncertainty in the form of fluctuations can promote ordering within the quantum system. Our findings shed light on the mechanisms and physical conditions leading to defect clustering under the action of quantum fluctuations, which can pave the way for improved quantum annealing systems. Furthermore, our experiments on entropic crystallization have revealed the remarkable influence of random fluctuations in inducing the emergence of ordered patterns in qubit systems, providing a solid foundation for future investigations into the manipulation and control of quantum states for improved computational performance.

Our embedding of nearly 2000 qubits in a superconducting quantum annealer to study large magnetic lattices demonstrates the effectiveness of quantum annealing in modeling the mechanisms that enhance correlations between quantum states. A detailed understanding of the physics of defect clustering can be obtained to guide the development of targeted strategies for minimizing errors and devising innovative approaches for interconnected qubit arrays. These findings may have implications for the design and organization of magnetic nanostructures as well as the engineering of quantum materials with desired properties.

## Data availability
Data supporting the findings of this paper are available from the corresponding author upon request.

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

## Acknowledgements
The authors acknowledge the contributions of the technical staff at D-Wave Quantum. The research presented in this paper was supported by the Laboratory Directed Research and Development program of Los Alamos National Laboratory under project numbers 20230546DR (A.L.-B.) and 20230049DR (A.S.). A.L.B., C.N., and A.S. work was carried out under the auspices of the U.S. DOE through Los Alamos National Laboratory, operated by Triad National Security, LLC (Contract No. 892333218NCA000001).

## Author contributions
A.L.B. and A.D.K. conducted the experiments on the D-Wave chip. C.N. provided a theoretical explanation of the phases and proposed the nonmonotonic transverse correlation that A.L.B. experimentally verified. A.S. proposed investigating frustrated deformed kagome lattices. A.L.B., C.N., and A.D.K. wrote the manuscript. All co-authors contributed to the final version of the manuscript.

## Competing interests
The authors declare no competing interests.
