## [Peer Review File · Nature Communications]

REVIEWER COMMENTS

Reviewer #1 (Remarks to the Author):

The manuscript titled “Quantum fluctuations drive nonmonotonic correlations in a qubit lattice” explores a realization of quantum Ising model on a Kagome lattice with superconducting qubits, where exchange interactions are tunable. The work studies the role of quantum fluctuations using a quench of the transverse field. In the coupled-chain regime, the nonmonotonic behavior of the transverse correlation is explained by an entropy preference by the domain wall locations.

It is a very impressive experiment with the large number of qubits and high tunability, which provides a perfect platform for the study of interacting quantum spin models. The different magnetic phases are clearly distinguishable and the quench sequence is well designed to address the connection between fluctuations and correlation. I judge that the work can have a significant impact in both areas of frustrated magnetism and quantum dynamics.

However, the interpretation related to the idea of order-by-disorder is very obscure throughout the manuscript.

1. The main reason for this is probably that it is not well explained why $1/J_L$ can be taken as an effective temperature.
2. A important piece of information is missing in the discussion of data in Fig.4: Is the annealing process of Γ unchanged while changing J and J_L ?
3. The general statement in the beginning of the introduction “The state of a disordered, ... based on their similar properties” is very confusing and misleading. How does this statement apply to the case of a uniform order parameter?
4. On page 5, “A first experimental result is that ... predicted ground states” seems unfinished and needs to be unpacked. What do the “excited states” refer to?

Lastly, I would like to invite the authors to consider the following alternative interpretation. Integrating out the fluctuating ϵ spin can induce an effective ferromagnetic coupling between the σ spins across the chains, which favor paired domain walls. Since this is a very natural way to approach the question, I think a discussion is necessary about how this is related to or different from the order-by-disorder idea.

Reviewer #3 (Remarks to the Author):

In the manuscript “Quantum fluctuations drive nonmonotonic correlations in a qubit lattice”, the authors presented experimental evidence that quantum fluctuations can lower the system energy by increasing entropy, similar to the order-by-disorder phenomenon. Central to their investigation is a non-trivial experimental setup. The authors prepared the ground state of a parameterized Hamiltonian by leveraging the quantum annealing (QA) algorithm. This was implemented on the D-Wave quantum hardware. The Hamiltonian is designed such that it is continuously transformed from a ferrimagnetic lattice to a paramagnetic layered structure by tuning the coupling strengths. The authors reported observing the non-zero and nonmonotonic behavior of the transverse spin correlation between each layer of spin chains, and attribute them to both quantum and thermal fluctuations.

With quantum hardware undergoing rapid advancements and improvements, the exploration of diverse avenues and applications they offer has become a task of paramount importance. The work is timely, well written and the results are clearly presented. However, the impact of the result is somewhat limited by a lack of discussion on any potential advantages of using quantum annealers over alternative methods. Specifically, the justification for choosing QA over strategies like directly cooling the system to the desired Hamiltonian's ground state, as showcased in Ref [12], remains unclear.

On a related note, it's also noteworthy that the results obtained from longer annealing time (100us) has comparable, if not stronger, “signal” than the ones obtained from shorter annealing time (1us). Because both of these two time scales are longer than the reported coherent time (see <https://www.nature.com/articles/s41586-023-05867-2> for example) of D-Wave devices, one can postulate that environmental cooling effects might be pivotal in this context. Consequently, this underscores the potential efficacy of alternative methods in similar scenarios.

It's my recommendation that the authors address these subjects more comprehensively before this research is considered for publication in Nature Communications.

One minor comment for the authors:

1. There's a point of potential ambiguity in the caption of Fig (1). The authors wrote: “Panels (c-e) show the Lieb lattice with anisotropic couplings J (solid gray lines, stronger), and J_L (dotted gray

lines, weaker).” This could be confusing for readers, as it may be interpreted that the "J couplings" themselves are stronger. It would be clearer to explicitly mention that the solid gray lines represent stronger couplings, irrespective of whether they are J or JL.

Authors' Response to Reviews of

Quantum fluctuations drive nonmonotonic correlations in a qubit lattice

By A. L. Bezanilla, A. King, C. Nisoli, A. Saxena
Nature Communications,

RC: Reviewers' Comment, AR: Authors' Response, Manuscript Text

1. Referee #1

RC: *The manuscript titled “Quantum fluctuations drive nonmonotonic correlations in a qubit lattice” explores a realization of quantum Ising model on a Kagome lattice with superconducting qubits, where exchange interactions are tunable. The work studies the role of quantum fluctuations using a quench of the transverse field. In the coupled-chain regime, the nonmonotonic behavior of the transverse correlation is explained by an entropy preference by the domain wall locations.*

It is a very impressive experiment with the large number of qubits and high tunability, which provides a perfect platform for the study of interacting quantum spin models. The different magnetic phases are clearly distinguishable and the quench sequence is well designed to address the connection between fluctuations and correlation. I judge that the work can have a significant impact in both areas of frustrated magnetism and quantum dynamics.

AR: We thank the reviewer for dedicating their time to review this manuscript and for providing valuable feedback, insightful observations, and positive recommendations.

RC: *However, the interpretation related to the idea of order-by-disorder is very obscure throughout the manuscript.*

1. The main reason for this is probably that it is not well explained why $1/J_L$ can be taken as an effective temperature.

AR: Thank you for pointing out that possible source of confusion. The environmental (the quantum processor unit) temperature is fixed to about 16 mK, namely it is effectively constant, so $1/J_L$ is simply a normalization of the Hamiltonian. Knowing that the ensemble is always described by a Boltzmann distribution, the statistical ensemble is controlled by $J/k_B T$. Therefore, an effective temperature of the system can be introduced, and controlled via modification of the coupling strength J between logical qubits, as ($T \sim 1/J$). Thus, high couplings correspond to low effective temperature, and vice versa. This assumption is commonly used and clarified in the manuscript as follows:

The coupling strength is subjected to at-will variations to mimic the effect of temperature, so that increasing J is equivalent to reducing the effect of thermal fluctuations. With discrete variations of the coupling strength ($J_L = 0.1..1.0$ in steps of 0.1) the system's effective temperature is mimicked, and it can be regarded as inversely proportional to the coupling strength between logical qubits ($T \sim 1/J_L$)

At high temperature, the interacting spins can be considered as decoupled since thermal fluctuations overwhelm the energy constraint imposed by the local field. A low temperature regime is equivalent to considering that thermal fluctuations are inefficient to induce a spin flip so that coupling between spins dominates the

interaction. Therefore, a weak coupling between individual qubits imposed throughout the system is considered as a high temperature state. Conversely, imposing the highest J_L between pairs of qubits mimics a state where thermal fluctuations are the lowest.

RC: *2. A important piece of information is missing in the discussion of data in Fig.4: Is the annealing process of Γ unchanged while changing J and J_L ?*

AR: Indeed, Γ is unchanged. At the outset of the annealing process, an initial quantum state is chosen to represent a known state of the problem Hamiltonian. Subsequently, the annealing process is reversed, effectively moving the system from this initial state to a quantum superposition state. Then, a specific value for the parameter Γ is set and held constant. At this constant value of Γ the process is paused and the system explores the landscape of possible states as part of the annealing process.

RC: *3. The general statement in the beginning of the introduction "The state of a disordered, ... based on their similar properties" is very confusing and misleading. How does this statement apply to the case of a uniform order parameter?*

AR: Thank you for bringing this to our attention. In the introduction, our aim is to offer a general and intuitive understanding of the concepts of order and disorder. We have revised the sentence to make it clearer and less confusing, as follows:

The state of a disordered system is defined by the even or uniform distribution of its constituent elements throughout space. Conversely, the concept of order pertains to the arrangement of these elements, where they tend to segregate or group together based on their similar properties.

RC: *4. On page 5, "A first experimental result is that ... predicted ground states" seems unfinished and needs to be unpacked. What do the "excited states" refer to?*

AR: We agree with the referee that the statement seems garbled. Also, "our first experimental results" are reported much earlier than that point. We have therefore removed the statement, and the paragraph ends with: *The system obeys the ice-rule at low T_{eff} .*

RC: *Lastly, I would like to invite the authors to consider the following alternative interpretation. Integrating out the fluctuating ϵ spin can induce an effective ferromagnetic coupling between the σ spins across the chains, which favor paired domain walls. Since this is a very natural way to approach the question, I think a discussion is necessary about how this is related to or different from the order-by-disorder idea.*

AR: The reviewer's observation is entirely accurate when considering the integration of thermal and even quantum fluctuations of an intermediate spin involving three spins that are coupled antiferromagnetically. This integration indeed results in an effective ferromagnetic coupling between the two remaining spins, and this coupling strength varies with temperature. While this interpretation was our initial hypothesis, it does not, however, provide a comprehensive explanation for the phenomenon. This is because the fluctuating spin is interconnected with two spins within each antiferromagnetic line (as illustrated in Figure 1e). Given that these two spins exhibit opposing orientations, their interactions with the fluctuating spin cancel each other. In simpler terms, the overall coupling between a fluctuating spin and an antiferromagnetic line is zero in the ground state.

However, when a defect occurs, such as a domain wall appearing in the antiferromagnetic order of the chain and aligning two σ spins in an upward direction, the net interaction between these aligned spins and the fluctuating Σ spin becomes non-zero. This interaction tends to align the fluctuating spin in a downward direction. Consequently, if there is a domain wall in the other line, that domain wall will also consist of a pair

of spins pointing upward. This alignment induces a transverse correlation between the two lines.

The key point of this argument is that there is no inherent energetic constraint requiring the presence of the second domain wall. In fact, relocating that domain wall beneath another Σ spin, one that does not have a domain wall above it, would not alter the system's energy but would instead fix the new Σ spin's position, thereby reducing entropy.

The three paragraphs following "*Figure 3 shows that the mechanism proceeds ...*" have been rephrased to:

Figure 3 illustrates that this mechanism proceeds through an entropic gain stemming from the correlation of domain walls. It is essential to note that in the ground state of the quasi-one-dimensional (Q1D) system, all σ -qubits lines (indicated by black circles) exhibit antiferromagnetic ordering, while the Σ qubits (indicated by yellow circles) have random orientations because their net coupling with the antiferromagnetic σ spins is zero. Consequently, in the ground state, there is no correlation among the antiferromagnetic lines. Excited states, on the other hand, involve the formation of domain walls on these lines, which separate domains with different antiferromagnetic orientations.

It is important to observe that a domain wall in an antiferromagnetic line possesses lower energy when located at the position of a Σ qubit, effectively locking it in an orientation opposite to that of the domain wall. In this scenario, the energy cost of creating a domain wall between two contiguous σ sites at both sides of a Σ spin is $\Delta E = 2(J_L - J)$. In contrast, placing the same domain wall away from a Σ spin (as depicted in Figure 3b) would result in a higher energy cost, $\Delta E = 2J_L$. It is therefore energetically advantageous for a domain wall to be adjacent to a Σ spin.

Next, consider two domain walls on two adjacent antiferromagnetic lines, with Σ spins in between. If these two domain walls lock onto the same Σ spin, they, and the domain they separate, adopt the same orientation, thereby contributing to a transverse correlation. However, there is no energy preference for having the two domain walls aligned by the same Σ spin. The energy remains the same if they lock onto two different Σ qubits. Nevertheless, there is an entropic advantage in having the two domain walls aligned because it allows one Σ spin to remain free to fluctuate, resulting in an energy gain of $\Delta S = \ln 2$. Consequently, we can deduce that, entropy promotes the alignment of domain walls in parallel chains and favors the existence of floppy spins, increasing the transverse correlation among these lines.

2. Referee #3

RC: *In the manuscript "Quantum fluctuations drive nonmonotonic correlations in a qubit lattice", the authors presented experimental evidence that quantum fluctuations can lower the system energy by increasing entropy, similar to the order-by-disorder phenomenon. Central to their investigation is a non-trivial experimental setup. The authors prepared the ground state of a parameterized Hamiltonian by leveraging the quantum annealing (QA) algorithm. This was implemented on the D-Wave quantum hardware. The Hamiltonian is designed such that it is continuously transformed from a ferrimagnetic lattice to a paramagnetic layered structure by tuning the coupling strengths. The authors reported observing the non-zero and nonmonotonic behavior of the transverse spin correlation between each layer of spin chains, and attribute them to both quantum and thermal fluctuations.*

With quantum hardware undergoing rapid advancements and improvements, the exploration of diverse avenues and applications they offer has become a task of paramount importance. The work is timely, well written and the results are clearly presented. However, the impact of the result is somewhat limited by

a lack of discussion on any potential advantages of using quantum annealers over alternative methods. Specifically, the justification for choosing QA over strategies like directly cooling the system to the desired Hamiltonian's ground state, as showcased in Ref [12], remains unclear.

AR: We thank the reviewer for their positive evaluation of our manuscript. Below, we address the comments and concerns.

We acknowledge that with the rapid advancements in quantum hardware, it has become increasingly crucial to explore various avenues and applications, including the advantages of using quantum annealers over alternative methods, such as the direct cooling method exemplified in that reference. It is important to note that, while spin ice systems are valuable for describing macroscopic physics, an important aspect of our research depends on leveraging the quantum effects facilitated by the qubits. The distinctive capabilities inherent to quantum annealers set them apart from alternative techniques, emphasizing the relevance of our approach. We take into account both thermal and quantum fluctuations in our analysis. The introduction of floppy spins contributes to an increased system entropy, a phenomenon effectively captured by the annealer to attain a state of lower free energy.

RC: ***On a related note, it's also noteworthy that the results obtained from longer annealing time ($100\mu s$) has comparable, if not stronger, "signal" than the ones obtained from shorter annealing time ($1\mu s$). Because both of these two time scales are longer than the reported coherent time (see <https://www.nature.com/articles/s41586-023-05867-2> for example) of D-Wave devices, one can postulate that environmental cooling effects might be pivotal in this context. Consequently, this underscores the potential efficacy of alternative methods in similar scenarios.***

It's my recommendation that the authors address these subjects more comprehensively before this research is considered for publication in Nature Communications.

AR:

A faster quenching process may result in a greater separation from the transverse field but, when considering a $1\mu s$ duration, we see that it is slow enough to approach a state akin to thermal equilibrium. If the correlations take a long time to evolve, it's very plausible that a $100\mu s$ could show more. Moreover, the fact that we can observe order-by-disorder phenomena at finite temperatures suggests that there is no need to be concerned about the coherence time of the qubits, as this observation belongs to an open-system scenario. Therefore, it is quite consistent with prior experiments to anticipate a stronger signal with a longer annealing period. Whether this effect is attributed to environmental cooling during the annealing process could seem uncertain, but the time frames involved indicate that cooling is not the primary factor. Even at a 100μ duration, the annealing time itself represents a small portion of the Quantum Processing Unit's duty cycle.

RC: ***One minor comment for the authors:***

1. There's a point of potential ambiguity in the caption of Fig (1). The authors wrote: "Panels (c-e) show the Lieb lattice with anisotropic couplings J (solid gray lines, stronger), and J_L (dotted gray lines, weaker)." This could be confusing for readers, as it may be interpreted that the " J couplings" themselves are stronger. It would be clearer to explicitly mention that the solid gray lines represent stronger couplings, irrespective of whether they are J or J_L .

AR: Thank you for bringing that up. For the sake of clarity we have modified that part of the caption to:

Panels (c-e) show the Lieb lattice with anisotropic couplings J (solid gray lines), and J_L (dotted gray lines). Solid gray lines represent stronger couplings.

REVIEWERS' COMMENTS

Reviewer #1 (Remarks to the Author):

I thank the authors for addressing my comments. I recommend the revised manuscript for publication.

Reviewer #3 (Remarks to the Author):

The authors have satisfactorily addressed my concerns. I now feel confident recommending the article for publication.